# Application of Combined Analyses of Stable Isotopes and Stomach Contents for Understanding Ontogenetic Niche Shifts in Silver Croaker (*Pennahia argentata*)

**DOI:** 10.3390/ijerph18084073

**Published:** 2021-04-13

**Authors:** Bohyung Choi, Won-Seok Kim, Chang Woo Ji, Min-Seob Kim, Ihn-Sil Kwak

**Affiliations:** 1Fisheries Science Institute, Chonnam National University, Yeosu 59626, Korea; chboh1982@chonnam.ac.kr (B.C.); jichangwoo@gmail.com (C.W.J.); 2Department of Ocean Integrated Science, Chonnam National University, Yeosu 59626, Korea; csktjr123@gmail.com; 3Department of Fundamental Environment Research, Environmental Measurement and Analysis Center, National Institute of Environmental Research, Incheon 22689, Korea; candyfrog77@gmail.com

**Keywords:** stable isotope analysis, stomach content analysis, ontogenetic niche shift, *Pennahia argentata*

## Abstract

Stable isotope analysis (SIA) and stomach content analysis (SCA) were conducted to understand ontogenetic niche shifts in silver croaker *Pennahia argentata* inhabiting the southern coastal waters of the Korean peninsula. Sampled *P. argentata* were classified into three groups based on their total length (TL; 60–80 mm TL, 80–120 mm TL, and 120–210 mm TL). Carbon isotope (δ^13^C) ratios were distinguishable, whereas nitrogen isotope (δ^15^N) ratios were not significantly different among size classes, and Standard Ellipse Area (SEA), estimated by δ^13^C and δ^15^N, was expanded with increasing TL from 0.2 ‰^2^ (60–80 mm TL) to 2.0 ‰^2^ (120–210 mm TL). SCA results showed variable contribution of dietary items to each size class. In particular, higher dietary contribution of Polychaeta to *P. argentata* of 80–120 mm TL than 120–210 mm TL mirrored variation in δ^13^C values of *P. argentata* in those size classes. Based on the combined analyses involving SIA and SCA, we concluded that *P. argentata* underwent ontogenetic niche shifts, particularly dietary shifts, with growth stages. Ontogenetic niche shifting is a representative survival strategy in fish, and, therefore, represents essential information for managing fisheries. The present study demonstrated applicability of combined SIA and SCA analyses, not only for dietary resource tracing, but also for ecological niche studies.

## 1. Introduction

The ecological niche, a fundamental concept in ecology, involves a variety of issues such as dietary resources, habitat information, and interactions among organisms [1]. Therefore, understanding the niche of organisms in an ecosystem has been considered as one of the most important tasks in ecology for several decades. One frequently applied tool for such study is stable isotope analysis (SIA), particularly using carbon and nitrogen isotope ratios. The minor difference in carbon isotope (δ^13^C) ratio between dietary resources and consumers provides diet information and the geographical habitat of consumers, whereas stepwise nitrogen isotopic enrichment along the food chain reflects trophic information regarding organisms [2,3]. Consequently, dual plots based on these two isotope ratios are used to illustrate food web structures of an ecosystem, with synthetically view of both dietary resources and trophic information, and, thus, the width of a polygon drawn by δ^13^C and δ^15^N values for a species is used as an ecological niche space (δ-space) [4,5]. The comparison of δ-space within and among species is used to understand ecological niches and interactions among organisms in an ecosystem [6,7,8]. However, complex information involving both δ^13^C and δ^15^N sometimes hinders us in identifying the key-factors driving ecological niche variation of organisms. For example, differences in δ^13^C values among consumers are generally derived from different dietary resources, but also indicate diverse habitats of organisms consuming homogeneous dietary resources [9,10]. The δ^15^N value of consumers generally indicates trophic hierarchy, but also can be varied to reflect that of basal organisms in a food web [11,12]. Therefore, variation in the δ-space implies diverse ecological interactions (i.e., competition or segregation involving habitat, diet, and different trophic status) and may lead to confusion in ecological interpretation.

Stomach content analysis (SCA) has traditionally been applied to identify consumer dietary items due to the provision of fine-resolution information at relatively low cost [13]. Moreover, SCA provides both quantitative and qualitative information concerning dietary items [14]. For instance, the weight of stomach contents provides quantitative information, and occurrence frequency indicates diet preference of a consumer, attributing quality of diet items for the consumer. However, only obtaining snapshot information and the requirement for enormous sample sizes are always regarded as inherent limitations to this approach [15,16]. Thus, recently, the combination of SCA and SIA offers a reinforcement tool for food web studies due to complementation of weaknesses inherent in each individual approach [13,15,17]. These combined analyses are also potentially applicable for niche identification studies. For instance, variations in δ-space involving both dietary and habitat changes can be more simply interpreted with information regarding dietary preferences obtained from SCA.

Silver croaker *Pennahia argentata* is a representative demersal fish distributed from east of Japan to the Indo Pacific, including the East China Sea and the Yellow Sea [18]. This species is regarded as commercially important, but catch limits have been gradually decreasing over recent years [19,20,21]. Despite its commercial importance, feeding strategies and dietary components of this species have rarely been reported and, moreover, information regarding ontogenetic diet shifting in this species is scarce. Dietary variability within a species has diverse purposes, such as nutritional requirements, morphological adaptation, and as a survival strategy in habitat competition [22]. Therefore, knowledge surrounding ontogenetic niche variation in a commercially important fish will be helpful in terms of managing marine resources production.

The purpose of the present study was to determine ontogenetic niche variation and diet changes in *P. argentata* caught in Gwangyang Bay, South Korea. Carbon and nitrogen stable isotope analysis was applied to determine niche widths for different groups of *P. argentata,* classified by total length, and their dietary information was obtained by SCA. This combined approach was used to explain the importance of dietary variations in determining *P. argentata* niche space.

## 2. Materials and Methods

### 2.1. Sampling and Study Sites

Gwanyang Bay is a semi-closed bay, and is connected with the Seomjin River. This bay is well known to have a high diversity of fish species due to the development of an estuary in the upper areas. Gwangyang Bay is geographically separated into four groups (estuary, main channel, outer bay, and inner bay), based on chemical and physical properties [23]. Sample collection was performed in September 2018 at three sites (St. 1, St. 2, and St. 3) in the main channel, which is characterized as exhibiting low metal concentrations, and at one site (St. 4) in the outer bay with high salinity (Figure 1). A small bottom-trawl net (length 8 m, width 8 m, mesh wing and body 3 cm, mesh liner 1 cm) was towed for 10 min at 1~2 knots at each sampling site. Seventy-eight samples were collected and immediately kept at −80 °C until delivery to a laboratory. Total length (TL) of each specimen was measured and classified into three groups based on their TL (60–80 mm TL, 26 individuals; 80–120 mm TL, 25 individuals; 120–210 mm TL, 27 individuals). The gut of each specimen was dissected and then preserved in 70% ethanol before examination. Forty-one samples were randomly selected, and dorsal muscle was dissected for SIA.

### 2.2. Stomach Content Analysis and Stable Isotope Analysis

Contents of the stomach were identified under a dissecting microscope (Olympus, SZX9, Tokyo, Japan) at a magnification of 20 to 100×, and classified into class levels. Because of diverse sizes, the unidentified contents were pooled and weighed together. Each identified content was weighed to the nearest 0.01 mg after removing excess moisture. The weight and frequency of unidentified contents were excluded in the estimation of relative abundance of diet items.

The extracted dorsal muscles were homogenized with a mortar and pestle after freeze-drying, and approximately 1 mg of each sample was sealed in a tin cap for carbon (C) and nitrogen (N) stable isotope analysis. Stable isotope analyses were performed using an isotope ratio mass spectrometer linked with an elemental analyzer (EA-IRMS, Isoprime, Manchester, UK). Samples for C stable isotope analysis generally required lipid removal; however, this process was omitted due to low and similar lipid contents in the samples [24]. The measured stable isotope ratios are expressed as ‘δ’ values by following Equation (1).
δ^13^C or δ^15^N = [(R_sample_/R_standard_) − 1] × 1000 (‰),(R = ^13^C/^12^C for carbon isotope, and ^15^N/^14^N for nitrogen isotope, respectively)(1)
where, vPDB (Vienna PeeDee Belemnite, International Atomic Energy Agency (IAEA)) and atmospheric N_2_ gas were used as standards for δ^13^C and δ^15^N, respectively. CH-3 (δ^13^C = −24.72 ± 0.1‰) and N-1 (δ^15^N = 0.4 ± 0.1‰) provided by the IAEA were analyzed after every five sample runs to check for analytical errors (1σ = ±0.1‰ for δ13C and 1σ = ±0.2‰ for δ15N, respectively).

### 2.3. Statistical Analysis

Significance of TL, δ^13^C, and δ^15^N of *P. argentata* across sampling zones defined by Kim et al. [23] was evaluated by one-way analysis of variance (ANOVA), and differences in δ^13^C, and δ^15^N among sizes classified for *P. argentata* were confirmed by two-way ANOVA. Tukey and Duncan’s multiple range methods were applied for post-hoc test using R (ver. 3.6.1).

Evaluation of isotopic niches for each TL-classified *P. argentata* was conducted by Stable Isotope Bayesian Ellipses in R (R package SIBER, Jackson et al. [5]), based on the δ^13^C, and δ^15^N values. The δ-space, Standard Ellipse Area (SEA), and overlap proportions were obtained using SIBER.

## 3. Results

### 3.1. P. argentata Spatial Variation

All *P. argentata* samples analyzed in this study demonstrated large ranges of δ^13^C (−16.6‰ to −13.1‰) and δ^15^N (11.8‰ to 15.5‰). However, both δ^13^C and δ^15^N of *P. argentata* sampled in the main channel and outer bay were not significantly different (*p* > 0.5) (Table 1). Therefore, isotopic niche evaluation in the present study did not involve consideration of spatial variation in isotope ratios of *P. argentata*.

### 3.2. Stable Isotope Ratios of Size-Classified P. argentata

δ^13^C values in *P. argentata* ranged from −15.6‰ to −13.7‰ in the 60–80 mm TL group, from −16.6‰ to −13.5‰ in the 80–120 mm TL group, and from −16.6‰ to −13.6‰ in the 120–210 mm TL cohort (Figure 2a). δ^13^C values in the 60–80 mm TL and 80–120 mm TL groups were not statistically different, whereas those of the 120–210 mm TL group were significantly different compared to the other groups (*p* < 0.05). δ^15^N values in *P. argentata* ranged from 11.8‰ to 15.5‰, with no statistical differences among size-fractionated groups (Figure 2b). However, the range of δ^15^N values was widest in 120–210 mm TL fish (11.8‰ to 15.5‰) and narrowest in the 60–80 mm TL group (13.4‰ to 15.2‰).

The *P. argentata* group comprising 80–120 mm was positioned relatively higher in terms of δ^13^C values in δ-space, whereas the 120–210 mm TL group showed greater depleted of their carbon isotopes (Figure 3a). The distinct δ-space in the 120–210 mm TL cohort resulted in little overlap with the 60–80 mm TL group (13.6%), while the overlapping δ-space between the 60–80 mm TL and 80–120 mm groups was 18.6%. SEA, which indicates niche width, was the smallest in the 60–80 mm TL group (0.4 ‰^2^), and increased with their TL (1.1 ‰^2^ for 80–120 mm TL, and 2.0 ‰^2^ for the 120–210 mm TL specimens, respectively), indicating expansion of niche width in this species accompanied by their growth (Figure 3b).

### 3.3. Stomach Contents of Size-Classified P. argentata

Stomach contents were examined in 50 specimens from 78 *P. argentata* samples; the remaining specimens (28 individuals) had empty guts. Among the size groups, specimens with empty guts were frequent in the 60–80 mm TL, while all gut samples had dietary contents in specimens of the 120–210 mm TL (Table 2). The amounts of unidentified contents followed size groups, with the highest in the 120–210 mm TL class, and the smallest in the 80–120 mm TL group. However, proportions of unidentified contents were higher in the 60–80 mm TL than in other size groups, and similar in the 80–120 mm TL and 120–210 mm TL specimens.

Although relative abundances were different, both total weight and frequency of diet contents significantly increased with fish size, indicating greater dietary consumption in greater TL specimens (Figure 4a,b). Indeed, large numbers of specimens with empty stomachs included *P. argentata* in the 60–80 mm TL group, resulting in simple diet information.

Dietary items in 60–80 mm TL group consisted only of Crustacea, Decapoda, and Isopoda, at similar frequencies (Figure 4b). However, relative weights of diet items were slightly higher in Isopoda (66.7%, Figure 4a). Meanwhile, more variable diet items, including Cephalopoda, Polychaeta, and Crustacea (Decapoda and Isopoda), were found in 80–120 mm TL class. Among the diet items, Polychaeta (43.8%) and Crustacea (37.5%) were present at high frequency, while Cephalopoda (12.5%) and Isopoda (6.25%) were detected at low frequency. Related abundances in biomass of diet items for 80–120 mm TL fish were also high in Polychaeta (44.5%) and Crustacea (37.3%), and low in Cephaloda (16.5%) and Isopoda (1.7%). Crustacea were the major dietary component of 120–210 mm TL group, as Decapoda (64.5%) and (25.8%) were detected at high frequency. Relative biomass of each diet item was also high in Decapoda (72.7%) and Isopoda (23.3%). Although the present study did not find this diet items in all stomach samples, previous studies have mentioned importance of Teleosts as diet items for *P. argentata*, [25,26]. In particular, Park et al. [26] demonstrated different diet contribution of teleosts for *P. argentata* between trawl and set net for sampling method. The ppresent study used trawl net for sampling, which may result in the absence of teleost in the stomach of *P. argentata*.

## 4. Discussion

The δ^13^C values of *P. argentata* analyzed in the present study exhibited a large range, indicating a broad ecological niche in this species. Ecological niche shifts within a species are generally related to their growth, and are, thus, considered as ontogenetic shifts [27,28]. Growth stages in fish are highly correlated with body size. For instance, Ju et al. [29] state that average body lengths in 0 year *P. argentata* samples are close to 80 mm, whereas one-year-old specimens predominantly grew up to approximately 120 mm TL. Among the *P. argentata* collected in the present study, specimens involving 120–210 mm TL can be assumed to be older than one year. The δ^13^C values in 120–210 mm TL specimens were significantly different compared to those in other size groups, with large ranges of δ^15^N values, while δ^13^C and δ^15^N values were not significantly different in 60–80 mm and 80–120 mm TL groups. These distributions in isotope values lead to distinct isotope niche widths among size-classified groups of *P. argentata*. However, broader SEAs in 80–120 mm TL fish (1.1) compared to those in 60–80 mm TL animals (0.4) indicated expansion of ecological niche with growth, and large overlap of ellipses in the 60–80 mm and 80–120 mm TL groups explained partial sharing of their habitat and diets. However, distinct δ-space in the 120–210 mm TL cohort indicated niche separation from other TL classes. Moreover, SEA in this class (2.0) was broader than in other classes, and clearly shows ontogenetic niche expansion with growth in this species. Indeed, distinctly low δ^13^C values (−15.7 ± 0.8‰) in fish larger than 170mmTL seems to offer further clear evidence for ontogenetic niche shifting. Unfortunately, our sample size was insufficient to evaluate niche shift within 120–210 mm TL fish, suggesting further future evaluation for niche shifts in adult *P. argentata*.

Ontogenetic niche shifts have two purposes: reducing predation risk through habitat shifts, and increasing growth through dietary shifts [22,30]. Applications involving SIA in ontogenetic niche studies have previously focused on diet shift scenarios. For instance, Post [22] discussed variable timing of transition from planktivory to piscivory in largemouth bass through nitrogen isotope analysis. Indeed, ontogenetic niche shifts relating to changes in food resources has been well demonstrated by SEA in diverse organisms (i.e., fur seal, Kernaléguen et al. [31]; krill, Zhu et al. [32]; fish, Krumsick and Fisher [33]). However, variable isotopic baselines among habitats are also attributed to wide ranges in isotope ratios. For instance, previous studies have reported higher δ^13^C values in species inhabiting inshore sites than in offshore species [34,35]. Therefore, geographical variation in isotope ratios within a given species has also been applied to understand ontogenetic shifts in habitats for diverse animals, including shark [36] and sea turtle [37]. Consequently, ontogenetic variation involving SEA can include shifts in both diet and habitat of a species. However, obtaining simultaneous information for both diet and habitat recorded in the isotope ratios of consumers hinders the ability to validate prevailing causes of niche shifts involving diet and habitat [11]. Thus, although distinct SEA between 80–120 mm and 120–210 mm TL *P. argentata* suggested ontogenetic niche shifts in this species, they did not inform us regarding the fundamental causes of habitat and dietary changes.

Meanwhile, SCA provides information concerning diet items for a species by optical resolution, and information on ontogenetic diet shifts, as well. Therefore, assuming that distinct SEA among the size-classified *P. argentata* is caused by diet shifts, *P. argentata* will have variable dietary items accompanying their body size. Unfortunately, resolution of our results involving SCA could not identify dietary items to species level due to large numbers of empty stomachs in the 60–80 mm TL class, and a high proportion of unidentifiable items (Table 2). Due to these issues, previous studies recommended large numbers of specimens for SCA to obtain more reliable information [15,16]. The number of samples for SCA in the present study (78 individuals) seems to be a relatively small number to obtain precise dietary information for *P. argentata*. Nevertheless, the large proportion of crustaceans identified in all sizes of *P. argentata,* as shown in our SCA results, is consistent with previous reports suggesting the importance of crustaceans as a major dietary component for this species by SCA [25] and SIA [38]. In addition, the contributions of Polychaeta as diet items for each size class of *P. argentata* were varied in a previous study [26]. Although size classification was slightly different in the present study, such previous reports support our findings in terms of the variable proportion of Polychaeta as a dietary resource among different size classes of *P. argentata*. Therefore, specific diet information was not involved in our SCA results; however, shifts in major dietary items among the size classes were certainly confirmed.

Furthermore, differences in diet composition between 80–120 mm and 120–210 mm TL of *P. argentata* were demonstrated by SCA and were well matched by our SEA results. Distinctively higher contributions of polychaeta as diet items in 80–120 mm TL group compared to 120–210 mm TL group seemed to result in separated δ-space between these classes. As δ^13^C values in benthic fish are higher than those in pelagic organisms in general [39,40], the higher δ^13^C value in δ-space of the 80–120 mm TL class than that in the 120–210 mm TL specimens, thus, reinforced our SCA results involving high benthic diet contributions in the 80–120 mm TL cohort. Meanwhile, crustacean remarkably contributes as diet item for 120–210 mm TL and also was important diet for 80–120 mm TL (approximately 40%). The considerable contribution of this common diet for these two groups explains partial overlap in δ-space between them. Therefore, our SCA results support that ontogenetic niche shifts in *P. argentata* were caused by dietary, rather than habitat, changes.

## 5. Conclusions

Understanding ontogenetic niche shifting is essential for management of fisheries, as it is one of the most representative survival strategies of organisms. The present study applied combined analyses of SIA and SCA to evaluate ontogenetic niche shift in a commercially important coastal fish, the Silver croaker *P. argentata*. Thus far, although combined analyses of SIA and SCA have been widely employed as powerful tools for tracing diet resources for marine consumers, this approach has not been used in ecological niche studies. The distinctive niche area between different fish size groups, as demonstrated by SIA, represents clear evidence of ontogenetic niche shifting. Moreover, our SCA results indicated that such an ontogenetic shift is strongly related to changes in dietary items concomitant with growth of *P. argentata*.

Consequently, the present study demonstrated the applicability of combined SIA and SCA analysis to obtain concrete information regarding ecological niches, and will help to further understand ecological strategies involving commercial marine fish.

## Figures and Tables

**Figure 1 ijerph-18-04073-f001:**
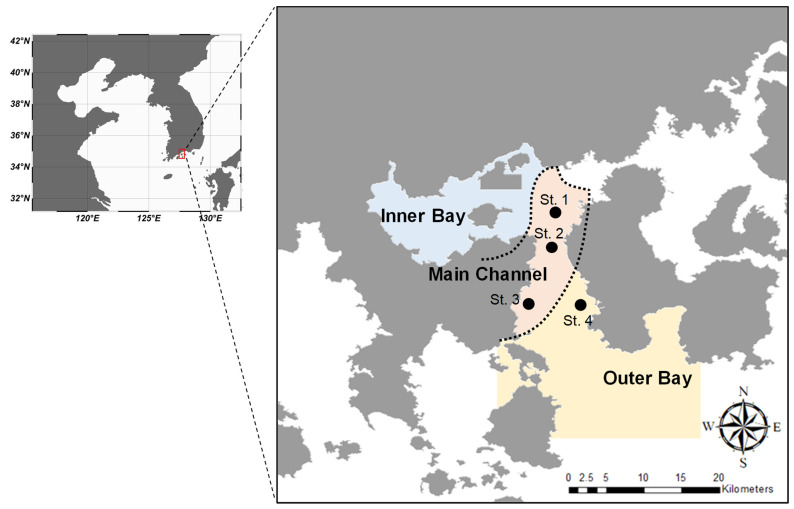
Map illustrating sampling sites in Gwangyang Bay. The boundaries in the bay are as defined by Kim et al. [23] 2019, based on physical and chemical properties.

**Figure 2 ijerph-18-04073-f002:**
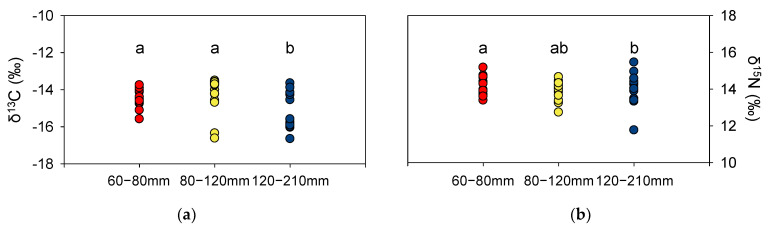
δ^13^C (**a**) and δ^15^N (**b**) values of *P. argentata* classified by their TL (mm). The letters indicate significant differences among size classes (*p* < 0.05 for δ^13^C, and *p* > 0.5 for δ^15^N).

**Figure 3 ijerph-18-04073-f003:**
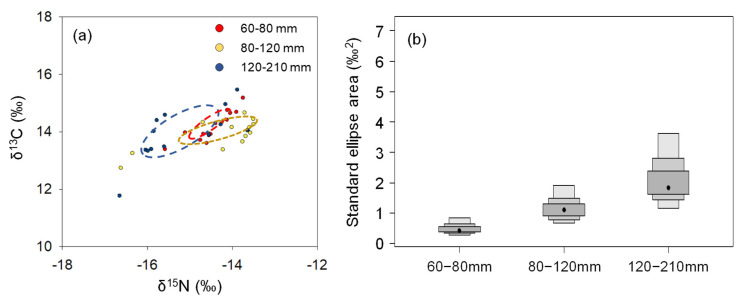
Isotopic niche of *P. argentata* classified by total length. (**a**) shows “δ-space” illustrated by dual plots of δ^13^C and δ^15^N values, and (**b**) indicates estimated standard ellipse area by SIBER for each class.

**Figure 4 ijerph-18-04073-f004:**
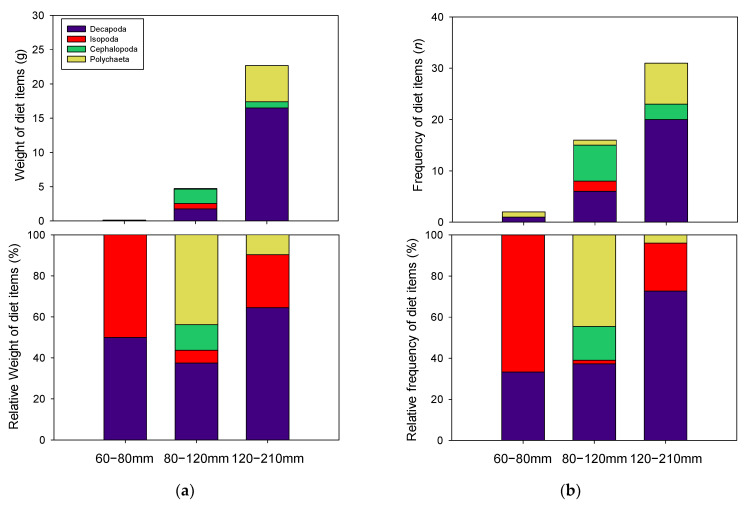
Stomach contents in each size class of *P. argentata*. (**a**) indicates contribution based on total weight and percentage of contents, and (**b**) indicates dietary contribution based on frequency and percentage of contents.

**Table 1 ijerph-18-04073-t001:** Spatial variation in carbon and nitrogen isotope ratios in *P. argentata* sampled at Gwangyang Bay. Main channel (St. 1, St. 2, St. 3 in Figure 1) and outer bay (St. 4) were separated by physical and chemical properties, as documented by Kim et al. (2019).

	Site	(n)	Avg. ± S.D.	*p*-Value ^1^
δ^13^C (‰)	main channel	25	−14.5 ± 1.1	0.2952
	outer bay	16	−14.8 ± 0.7	
δ^15^N (‰)	main channel	25	14.0 ± 0.7	0.520
	outer bay	16	14.1 ± 0.5	
Total Length (TL, mm)	main channel	55	112.7 ± 46.0	0.4639
	outer bay	22	122.9 ± 57.6	

^1^ Statistical significance was evaluated by one-way ANOVA.

**Table 2 ijerph-18-04073-t002:** Information regarding empty and unidentified stomach contents in *P. argentata* classified by total length.

Size Classes of *P. argentata*	60–80 mm	80–120 mm	120–210 mm
No. of examined specimens (n)	26	25	27
No. of empty stomachs (n)	19	9	0
% of empty stomachs (%)	73.1	36	0
Total Weight of stomach contents (g)	0.3	5.5	31.6
Weight of unidentified contents (g)	0.2	0.8	8.9
% of unidentified contents (%)	62.5	14.2	28.1

## Data Availability

The data presented in this study are available in “Application of combined analyses of stable isotopes and stomach contents for understanding ontogenetic niche shifts in Silver croaker (*Pennahia argentata*)”.

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
