# Peer review of "Application of Combined Analyses of Stable Isotopes and Stomach Contents for Understanding Ontogenetic Niche Shifts in Silver Croaker (Pennahia argentata)"

_ijerph, 2021, doi:10.3390/ijerph18084073_

Round 1
Reviewer 1 Report
Application of combined analyses of stable isotopes and stom-2 ach contents for understanding ontogenetic niche shifts in Silver croaker (Pennahia argentata)
Manuscript: IJERPH-1149202
This manuscript reports a discussion on the use of combined analyses of stable isotope analysis (SIA) and stomach content analysis (SCA) as an approach to studying ontogenetic niche shift of the commercially coastal fish “Pennahia argentata” from Gwangyang Bay (South Korea). The achieved results indicated that the ecological niches shift is related to changes in dietary needs/resources. The study intends to contribute to understanding ecological strategies related to commercial marine fish.
I consider the article is within the research interests of IJERPH and adequate to be published in this scientific journal and recommend its acceptance for publication.
The manuscript is, in general, well structured and the presentation of data and discussion is clear and founded with statistical analysis of the data. The major findings are presented in an objective mode.
The graphical presentation of the data should be improved, to catch better the reader’s interest.
Minor comments
Revise and standardize the designation - Silver croaker (Pennahia argentata) / silver croaker Pennahia argentata
Line 86 needs revision.
Author Response
We thank the Reviewer #1 for the positive comments and detailed editing on our manuscript. The manuscript has been revised to address the Reviewer #1’s concerns and suggestions.
Please see the attached file for revision

Reviewer 2 Report
The manuscript entitled ‘’Application of combined analyses of stable isotopes and stomach contents for understanding ontogenetic niche shifts in Silver croaker (Pennahia argentata)’’ presents an investigation on the ecological niche of a commercial fish species. This kind of research is necessary to understand species ecology and has important application for fisheries management and environmental protection. The study applied both, stomach content analyses and stable isopods methods which I believe is the correct approach to unravel the complex trophic relationships in marine food webs. In addition, ontogenetic shifts are also envisaged, providing clue-information about habitat uses during the lifespan on the species. Although mainly descriptive, I think this is an in-depth work that deserves publication.
Also, the manuscript is very clear and well written. I hardly could find a flaw after several revisions. After all, my only concern is that I think this kind of manuscript would better fit in a journal such as Fishes, Biology or Marine sciences and engineering.
Author Response
Response:
We thank the Reviewer #2 for the positive comments. The aim of “IJERPH” covers wide disciplines including biology and ecology. We believe that our manuscript will contribute to understand ecological role of silver croaker, and thus sure that purpose is suit for “IJERPH”.
Reviewer 3 Report
The study reported uses stable isotope analysis (SIA) and stomach content analysis (SCA) to assess ontogenetic niche shifts in silver croaker Pennahia argentata inhabiting southern coastal waters of the Korean peninsula.
The study uses 78 specimens that were collected with a bottom-trawl and consider three size-classes (60-80 mm, 80-120 mm and 120-210 mm in TL). Overall, the authors used appropriate sampling and analytical methods in the reported study. Although the sample size is somewhat small given the study objectives (only 50 fish specimens had food in their stomach contents) and only broad food categories are used, the authors try to deal with those constrains in the ms. discussion. Nevertheless they remain weak points in the study.
The major result of the study is the confirmation that P. argentata underwent dietary shifts with size in the study area. However, dietary shifts with size have already been described for the species elsewhere (Huh et al 2018). Despite these limitations, the combined use of SIA and SCA increases the relative interest of the study, although I have some doubts if that interest could justify publication in International Journal of Environmental Research and Public Health (IJERPH). Finally, I also have some doubts wheter the topic of the study is within the aims and scope of IJERPH.
The ms. is generally well written, but some sentences need to be rewritten.
Below are some specific comments that could help the authors in an eventual revision of the ms.
Lines 70-71 “This species is regarded as commercially important, but catchment limits have been gradually decreased over recent years”. Please rewrite this sentence, e.g. This species is regarded as commercially important, but catch limits have been gradually decreasing over recent years.
Line 86 “Gwangyang Bay is a semi-closed bay known to be large diversity of fish species (…).” Please rephrase this sentence, e.g. Gwangyang Bay is a semi-closed bay known to harbour a high diversity of fish species (…).
Table 1. Why are the fish samples from the outer bay always larger than the ones from the main channel, although 3 sites were sampled in the main channel and only 1 in the outer bay? Is the species more abundant in the outer bay?
Lines 179-180 “Although relative abundances were different, both total weight and total frequency of diets in P. argentata were significantly increased with fish TL (…)” Please rewrite this sentence, e.g. Although relative abundances were different, the total weight of diet contents significantly increased with fish size (…)”
Lines 216-217 “Indeed, distinctly low δ13C values (−15.7‰ ± 216 0.8‰) in < 170 mm TL fish seems to offer further clear (…)”. This size class (i.e. fish < 170 mm) is not considered in the study.
Given that fish are an importante prey item for the studied species elsewhere and that the sampling methods used could bias the dietary results (Park et al 2020), I would like to have some mention about that in the ms.
References cited
Huh SH, Choi HC and Park JM (2018) Feeding Relationship between Co-occurring Silver Croaker (Pennahia argentata) and Japanese Sillago (Sillago japonica) in the Nakdong River Estuary, Korea. Korean Journal of Ichthyology 30: 224-231.
Park JM, Kwak SN, Lee WC (2020) Dietary study using set-nets produces bias in prey choice of fish: A case of three coastal fishes inhabiting southern Korean waters. Journal of Sea Research 157 (2020) 101846,
Author Response
We greatly appreciate the reviewer #3 for his/her valuable comments on our manuscript.
We agree that dietary shifts with size in P. argentata have been reported via gut content analysis. However, our isotope data enhanced our knowledge on ontogenetic niche shift in P. argentata. Not only diet shift, SIA data also is telling us expansion of niche width with their growth. We believe that combination analysis of SIA and SCA to ecological niche study brought us more ecological information on this species. The aim of “IJERPH” covers wide disciplines including biology and ecology. We believe that our manuscript will contribute to understand ecological role of silver croaker, and thus sure that purpose is suit for “IJERPH”.
Please see the attached file for the replies of the recommandation from reviewer #3
